# *ScATG8* Gene Cloned from Desert Moss *Syntrichia caninervis* Exhibits Multiple Stress Tolerance

**DOI:** 10.3390/plants13010059

**Published:** 2023-12-23

**Authors:** Ting Cao, Yakupjan Haxim, Xiujin Liu, Qilin Yang, Amangul Hawar, Abdul Waheed, Xiaoshuang Li, Daoyuan Zhang

**Affiliations:** 1State Key Laboratory of Desert and Oasis Ecology, Key Laboratory of Ecological Safety and Sustainable Development in Arid Lands, Xinjiang Institute of Ecology and Geography, Chinese Academy of Sciences, Urumqi 830011, China; caoting21@mails.ucas.ac.cn (T.C.); yakoo@ms.xjb.ac.cn (Y.H.); liuxiujin@ms.xjb.ac.cn (X.L.); yangqilin22@mails.ucas.ac.cn (Q.Y.); amanguli@ms.xjb.ac.cn (A.H.); drwaheed@ms.xjb.ac.cn (A.W.); lixs@ms.xjb.ac.cn (X.L.); 2University of Chinese Academy of Sciences, Beijing 100049, China; 3Xinjiang Key Laboratory of Conservation and Utilization of Plant Gene Resources, Xinjiang Institute of Ecology and Geography, Chinese Academy of Sciences, Urumqi 830011, China; 4Turpan Eremophytes Botanical Garden, Chinese Academy of Sciences, Turpan 838008, China

**Keywords:** desiccation, abiotic stress, ATG8, *Syntrichia caninervis*

## Abstract

*Syntrichia caninervis* is the dominant species of biological soil crust in the desert, including the Gurbantunggut Desert in China. It is widely distributed in drylands and considered to be a new model of vegetative desiccation tolerance moss. Here, we cloned an *ATG8* gene from *S. caninervis* and confirmed its function under multiple abiotic stresses, both in situ and in *Physcomitrium patens*. The results showed that the *ScATG8* gene encoded a protein with a highly conserved ATG8 functional domain. *ScATG8* gene was increasingly expressed under different abiotic stresses. Under desiccation stress, the overexpression of *ScATG8* enhanced the tolerance of *S. caninervis* and its ability to scavenge ROS. In addition, *ScATG8* overexpression promoted the growth of *P. patens* under multiple stress conditions. Thus, *ScATG8* may be a multifunctional gene, and it plays a critical role in the survival of *S. caninervis* under various abiotic stresses. Our results provide new insights into the function of *ATG8* in enabling desiccation tolerance and open up more possibilities for subsequent plant molecular breeding and the mining of the resistance genes of *S. caninervis* and other moss species.

## 1. Introduction

ATG8 is a ubiquitin-like protein that plays a crucial role in autophagy [1]. During autophagy, ATG8 binds to phosphatidylethanolamine (PE) and plays an important role in forming autophagic vesicles [2]. In plants, *ATG8* has been found to be involved in a wide range of biotic and abiotic stresses [3]. *TdATG8* cloned from wild emmer wheat was able to positively regulate osmotic and drought stress in transgenic yeast [4], and it also demonstrated a positive response in wheat affected by high salinity, drought, low temperature/darkness, and nitrogen deficiency [5]. Recent studies have shown that overexpression of *TaATG8a* enhances salt tolerance in wheat seedlings; reduces photosystem II damage and the accumulation of reactive oxygen species; enhances the activities of superoxide dismutase, peroxidase, and catalase; and inhibits programmed cell death (PCD) under salt stress [6]. Pear *PbrATG8* showed upregulation under drought and salt stress [7]. Interestingly, it was found that both the silencing and overexpression of the pepper *CaATG8c* gene made plants more sensitive to heat and salt stress [8]. In addition to participating in abiotic stress responses, it can recognize virulence proteins of pathogens such as viruses [9] and fungi [10,11].

In nature, desiccation is a more extreme abiotic stress than drought stress, meaning that it poses a great threat to plants. Plants were considered to be desiccation tolerant if they could survive leaf water potentials at drying to −100 MPa. Plants that are able to regain vitality despite losing 95% of protoplasmic water from nutritive tissues or dehydrating to a water content of 0.1 g H_2_O/g dry weight are called desiccation-tolerant plants [12,13,14]. Desiccation-tolerant plants are able to adapt to dry environmental conditions through a variety of morphological, physiological, and molecular mechanisms, with plants in a state of water deficit first delaying growth [15].

*Syntrichia caninervis*, as a typical desiccation-tolerant plant, is widely distributed in the arid zones of Central Asia and the Mojave Desert of North America, and it is a dominant species in the biological soil crust of the Gurbantunggut Desert in Xinjiang, China [16]. It was shown that *S. caninervis* shares 44% of its protein family with *Arabidopsis thaliana*, and, thus, *S. caninervis* may have unique tolerance mechanisms, in addition to those shared with other species [17]. *S. caninervis* is able to lose more than 90% of its protoplasmic water under desiccation stress and subsequently rehydrate to regain physiological activity [18], making it one of the few plants that has undergone evolution to develop vegetative desiccation tolerance [19]. This characteristic also provides *S. caninervis* with a stronger adaptive ability for survival in desert areas, which is important for maintaining the stability of desert ecosystems and drought research.

*P. patens* is a powerful non-seed plant model system that has made fundamental contributions to diverse fields, including evolutionary developmental and cell biology [20]. *P. patens* is relatively more sensitive to desiccation stress than *S. caninervis*, and *P. patens* did not survive water potentials < −13 MPa [21], while *S. caninervis* withstood water potentials of −540 MPa [22]. *P. patens* has a short life history cycle, it is easy to cultivate, and the transgenic plants are easy to analyze and can be screened and experimented on in a short period of time [23]. Thus, these characteristics make it a good material for studying desiccation tolerance gene function.

Recently, some progress has been made in the cloning and functional study of resistance genes in the *S. caninervis*, such as *ScDREB10* [18], *ScALDH21* [24], and *ScAPD1-like* [25]. Here, we investigated the transcriptional regulation of the *ScATG8* gene and its function during abiotic stresses. *ScATG8* was upregulated under desiccation, salt, and heat stress. The overexpression of the *ScATG8* gene in *P. patens* promoted growth under different stress conditions. Our results extend the knowledge of the *ATG8* gene and indicate its novel role in response to multiple abiotic stresses.

## 2. Results

### 2.1. Cloning and Phylogenetic Analysis of ScATG8

To identify the *ATG8* genes in *S. caninervis*, a Hidden Markov Model (HMM) profile of the Atg8 domain (PF02991) was used as a BLAST query against the *S. caninervis* genome (GenBank assembly: GCA_016097705.1). As a result, we identified the *ATG8* homologous (E-value 1.00 × 10^−51^) gene and named it *ScATG8*. The *ScATG8* gene is located on chromosome 7 (Figure 1A), and it is 1078 bp in length with four exons (Figure 1A,B). Sequence analysis revealed that the full-length CDS sequence of *ScATG8* consisted of 372 nucleotides, which encoded a 123-amino-acid peptide (Appendix A) with a putative molecular weight of 13.78 kDa. The phylogenetic analysis clearly indicates that ScATG8 protein has a close genetic relationship with PpATG8 (Figure 1C). Further multiple sequence alignment showed that ScATG8 shared a high sequence identity with ATG8 proteins derived from different plant species. SMART domain architecture analysis showed that ScATG8 and other ATG proteins have an ATG8-based conserved functional domain (Figure 1D). The 3D structure analysis revealed that ScATG8 shared a similar ubiquitin-like fold with two additional amino-terminal α helices or C-terminal glycine cleavage sites with its orthologs from Arabidopsis and yeast (Figure 1E). These results showed that ATG8 protein is structurally conserved in *S. caninervis*.

### 2.2. Expression Pattern of the ScATG8 Gene under Different Abiotic Stresses

The expression pattern of the *ScATG8* gene under different abiotic stresses showed that there was a significantly change under dehydration–rehydration (Figure 2A), salt (Figure 2B), cold (Figure 2C), and heat (Figure 2D) stresses. The expression of *ScATG8* was significantly increased during the dehydration period and reached its peak (14-fold increase) at the end of dehydration (48 h) (Figure 2A), and then it gradually decreased after rehydration. *ScATG8* was significantly elevated 2-fold at 2 h after salt stress (Figure 2B) treatment and began to decline after peaking at 8 h. *ScATG8* rapidly responded to cold stress (Figure 2C) and early heat stress (Figure 2D) but then declined to the control levels.

### 2.3. Subcellular Localization of ScATG8

To determine the localization of ScATG8 in cells, we constructed the pCAMBIA1300-CFP-ScATG8 plant expression vector and transiently expressed it in *S. caninervis*. The cells were observed under a confocal microscope after 7 days of agroinfiltration. Compared to control cells, CFP-ScATG8-expressed cells displayed the autophagic body (Figure 3A). In the *N. benthamina* the CFP-ScATG8 fusion protein was also localized in autophagic vesicles and showed Brownian motion in the cytoplasm (Figure 3B). These results suggest that ScATG8 protein was functionally conserved and involved in autophagosome biogenesis.

### 2.4. Function of ScATG8 during Dehydration–Rehydration

To investigate the function of the *ScATG8* gene, we generated *ScATG8* gene overexpressed (ScATG8^OE^) (Figure 4A) and *ScATG8* gene silenced (ScATG8^RNAi^) lines (Figure 4B) in plants. The measurement of the maximum photochemical efficiency (Fv/Fm) showed that the overexpression of *ScATG8* leads to rapid decreases in Fv/Fm values during the dehydration period but accelerates Fv/Fm recovery in the rehydration period (Figure 4C). ScATG8^RNAi^ plants showed no significant change in Fv/Fm (Figure 4D). Absolute water content (AWC) was also examined during the dehydration–rehydration process. The results showed that ScATG8^OE^ and ScATG8^RANi^ plants rapidly lost most of their water within 24 h during the dehydration process and almost completely dried after 48 h (Figure 4E,F). Interestingly, after rehydration AWC dramatically increased and completely recovered in 0.5 h in ScATG8^OE^, while AWC in ScATG8^RANi^ plants was always below the initial level. No morphological changes were observed in ScATG8^OE^ plants compared to the control (empty vector). During dehydration, ScATG8^RANi^ plants dehydrated in a slightly slower manner, and their surfaces were dried after 24 h (Figure 4G).

### 2.5. Overexpression of ScATG8 Promotes ROS Scavenging in S. caninervis under Dehydration–Rehydration

To investigate the role of *ScATG8* in the ROS clearance system, the H_2_O_2_ and O_2_^−^ levels in the ScATG8^OE^ and ScATG8^RNAi^ lines were examined using DAB and NBT staining, respectively. The results showed that the overexpression of *ScATG8* leads to a decrease in the H_2_O_2_ level, but H_2_O_2_ was accumulated in the ScATG8^RNAi^ lines during the rehydration period (Figure 5A). Similarly, the O_2_^−^ level in ScATG8^OE^ plants was lower than the empty vector control and ScATG8^RNAi^ lines (Figure 5B). These results suggested that *ScATG8* contributes to ROS clearance in *S. caninervis*.

### 2.6. ScATG8 Overexpression Promotes Cold, Heat, and Salt Tolerance in Transgenic Physcomitrium Patens

In order to study the function of *ScATG8* under different abiotic stresses, *ScATG8* transgenic *P. patens* was generated, and three transgenic lines (ScATG8^OE^-L2, ScATG8^OE^-L3, and ScATG8^OE^-L6) were randomly selected for further study (Figure 6A,B). There was no significant morphological and phenotypical difference between three transgenic lines and the wild type (WT) under normal growth conditions (Figure 6C). Furthermore, CFP-ScATG8 fusion proteins were observed and localized to autophagic vesicles in transgenic lines compared to the wild type (Figure 6D). Transgenic lines were less sensitive in earlier stages of desiccation treatment compared to the wild type (Figure 7A). A large number of wild type plants died after desiccation treatment on the first day, whereas transgenic plants survived until the third day of stress treatment. However, by the seventh day, all plants had died (Figure 7A,B).

The heat, cold, and salt stresses were not lethal to transgenic or wild-type *P. patens* but affected its growth. The protenemal area, protonemal length, and protonema growth rate were measured at different time points after stress treatment. Under heat, cold, and salt stress, transgenic lines grow better than wild-type plants with larger protonemal areas (Figure 7C–H), protonemal length (Appendix A), and protonema growth rate (Appendix A). In summary, the overexpression of *ScATG8* promotes desiccation, heat, cold, and salt stress tolerance in *P. patens*.

## 3. Discussion

Autophagy, as a cytolytic metabolic process, plays a key role in restoring homeostasis in eukaryotic cells. ATG8 is key protein that participates in various stages of the autophagic process. Here, we identified a single *ATG8* gene from *S. caninervis*, a typical desiccation tolerance moss that revives from extreme water loss. It is well characterized that ATG4, a cysteine proteinase, cleaved the C-terminus of newly synthesized ATG8 and exposed the C-terminal glycine residue [2]. Previous studies have indicated that ATG8 proteins conserved among eukaryotes contains an N-terminal helical domain and C-terminal ubiquitin domain [3]. Like other eukaryotic ATG8 proteins, the ScATG8 also contained C-terminal glycine residue and has a N-terminal helical domain and a C-terminal ubiquitin domain (Figure 1). It is evident from these characteristics that ScATG8 is a highly conserved structural protein. There is only one *ATG8* gene in yeast and green algae [3]. In order to adapt to an adverse and complex environment, green plants expanded the *ATG8* gene family via multiple whole-genome duplications [28]. Notably, the *S. caninervis* genome only contained a single *ATG8* gene, implying that this single *ATG8* gene serves multiple functions. The gene expression of *ScATG8* under different abiotic stresses demonstrated that *ScATG8* was involved in diverse biological process.

A recent study showed that desiccation stress triggers autophagy in resurrection plants and inhibits programmed cell death (PCD), maintaining the tissue viability of desiccation-tolerant plants in the desiccation state and exerting a protective effect on plants [29]. However, in *P. patens*, autophagy instead results in cell death after desiccation [30]. *S. caninervis* is a desiccation-tolerant species of moss. During dehydration–rehydration, the transient overexpression of *ScATG8* enhanced ROS scavenging and protected *S. caninervis* cells from oxidative damage (Figure 4 and Figure 5). This suggests that *ScATG8* can scavenge ROS through cellular autophagy in *S. caninervis* under desiccation stress. Meanwhile, the analysis of photosynthesis revealed that Fv/Fm decreased faster during dehydration and recovered faster after rehydration in ScATG8^OE^ plants. However, the specific mechanism of *ScATG8* involvement in the dehydration–rehydration of *S. caninervis* is still unclear. Further research is needed to uncover the role of autophagy during the dehydration–rehydration of this kind of *S. caninervis*. In this study, we also generated *ScATG8* transgenic *P. patens*. The heterologous expression of *ScATG8* gene enhanced the desiccation tolerance of *P. patens*. Interestingly, the overexpression of *ScATG8* in *P. patens* facilitated the growth and development of protonema under heat, cold, and salt stress, suggesting that *ScATG8* may mediate these abiotic stress responses through autophagy. It was also shown that overexpressing *OsATG8b* promoted the growth and development of Arabidopsis by increasing nitrogen use efficiency [31]. However, whether the overexpression of *ScATG8* affects autophagy and whether it is ultimately involved in dehydration–rehydration stress tolerance in *S. caninervis* through autophagy is unknown.

On the basis of the experimental results, we concluded that *ScATG8*, a cellular autophagy-related gene in *S. caninervis*, can play a role in dehydration–rehydration stress, salt stress, and heat stress. The measurement of Fv/Fm revealed that ScATG8^OE^ plants showed better photosynthetic recovery after desiccation, and their water content was able to recover more quickly after rehydration. Conversely, ScATG8^RNAi^ plants had worse photosynthetic recovery values. In addition, *ScATG8* overexpression enhanced the scavenging of ROS. The stable transformation of the ScATG8^OE^ vector into *P. patens* and subjecting it to desiccation stress, salt stress, cold stress, and heat stress showed that ScATG8^OE^ transgenic lines grew better than WT. As a result, we have gained a better understanding of the physiological responses of plants under extreme stress, which will improve the plants’ breeding methods and open up opportunities to exploit the high-quality germplasm resources of *S. caninervis* and other mosses.

## 4. Materials and Methods

### 4.1. Sequence and Phylogenetic Analysis of the ScATG8 Gene

The amino acid sequence ATG8 of 13 different species were obtained from NCBI (http://www.ncbi.nlm.nih.gov/), (https://www.arabidopsis.org/), (https://www.hornworts.uzh.ch/en/download.html), and the Chlamydomonas database (http://www.chlamy.org/chlamydb.html). All these databases were accessed on 1 May 2023. The phylogenetic tree was constructed using MEGA7 software [32] using the neighbor-joining method with 1000 replicates of the Bootstrap method.

### 4.2. Plant Growth

*S. caninervis* were collected from Gurbantunggut Desert in Xinjiang, China (44°36′ N, 88°15′ E). For the experiments, the *S. caninervis* samples were washed with distilled water and then, after being removed, the rhizoids the gametophytes were re-placed onto a peat pellet (JiffyCorp., Winnipeg, MB, Canada) and grown under 14 h light/8 h dark at 24 °C, with a light intensity of 50 μmol m^−2^ s^−1^ [33].

### 4.3. Stress Treatment for S. caninervis and Transgenic Physcomitrium Patens

Dried *S. caninervis* gametophytes were rehydrated with ultrapure water for 24 h to reach a fully hydrated state, and then [34] part of the root system was removed and prepared for the experiment. The dehydration–rehydration treatment was performed by slow drying the samples in a drying chamber (RH = 66–67%) containing saturated sodium nitrite solution, and the samples were collected at 0, 6, 24, and 48 h, respectively [35]. Subsequently, the fully dehydrated gametophytes were transferred into petri dishes for rehydration, and the samples were taken at 0.5, 6, and 24 h, respectively. Salt stress, cold stress, and heat stress were applied by incubating *S. caninervis* gametophytes in 250 mM NaCl solution [25] for 48 h at −4 °C for 24 h and 40 °C for 3 h. Samples of salt stress and heat stress were taken at 0, 0.5, 2, 4, 8, 24, and 48 h. Samples of cold stress were taken at 0, 1, 3, 8, 12, and 24 h. The stress treatment of ScATG8^OE^ transgenic *P. patens* material selection was performed on protonema. The desiccation stress treatment was performed by transferring *P. patens* protonema onto filter paper and placing it in sealed Petri dishes containing silica gel, rehydrating it immediately after 6 h of desiccation, and incubating it using Knop’s agar medium; cold-stressing it at −4 °C for 24 h; and heat-stressing it at 40 °C for 3 h before placing it in the normal conditions and culturing it under normal conditions (16 h/8 h photoperiod, 15 °C/14 °C day/night temperature, 150 μmol m^−2^ s^−1^ light intensity); the samples were photographed and observed on days 0, 1, 3, and 7. For salt stress treatment, protonema was placed in Knop’s agar medium with 250 mM NaCl for incubation, photographed, and observed after 0, 3, 7, and 15 days.

### 4.4. RNA Extraction and RT-qPCR Analysis

Total RNA was extracted from 0.1 g of the collected plant samples, and this experiment was carried out according to the instructions of E.Z.N.A.^®^ Plant RNA Kit (Omega Bio-tek, Norcross, GA, USA). The cDNA was obtained by using the PrimeScript™ RT reagent Kit with gDNA Eraser (Perfect Real Time, Takara, Kyoto, Japan) to reverse-transcribe RNA into a total of 1 μg of cDNA. The procedure for reverse transcription was as follows: the reaction solution was 2 μL of 5× gDNA Eraser Buffer, 1 μL of gDNA Eraser, and total RNA and RNase Free dH_2_O, totaling 10 μL, which was placed in a PCR machine and reacted at 42 °C for 2 min, and then 1 μL of PrimeScript RT Enzyme Mix I was added to the reaction solution. Next, 1 μL RT Primer Mix, 4 μL 5× PrimeScript Buffer 2 (for Real Time), and 4 μL RNase Free dH_2_O were added to the reaction solution at 37 °C for 15 min and 85 °C for 5 s. All cDNA was stored at −20 °C. The cDNA was used as a template for RT-qPCR after 5-fold dilution. RT-qPCR was performed using TB Green^®^ Premix Ex Taq™ II (Tli RNaseH Plus, Takara, Kyoto, Japan); according to the instruction manual, we applied the CFX96 Real-Time PCR Detection System method of operation. The RT-qPCR program involved pre-denaturation at 95 °C for 30 s, followed by 40 cycles of 95 °C for 5 s, 58 °C for 15 s, and 72 °C for 15 s, using *ScTubulin2* as the internal reference gene. Relative gene levels were calculated using 0 h as a control group, and three separate biological replicates were performed for each type of sample, and the data were analyzed according to the 2^−ΔΔCt^ calculation method [27]. Transcript abundance data were collected using the Bio-Rad CFX Manager. The one-way ANOVA test analysis was performed for RT-qPCR data analysis using SPSS 25 software. Sequence specific primers are listed in Appendix A.

### 4.5. Plasmid Construction

The full CDS sequence of *ScATG8* (Appendix A) was cloned through PCR, and then the ScATG8-positive plasmid was digested with the restriction enzymes HindIII and XbaI and subcloned into the pCAMBIA1300-CFP binary vector for overexpression in *S. caninervis*. The sense sequence of the *ScATG8* gene (Appendix A) was digested with the restriction enzymes NcoI and SwaI and silenced in the plant expression vector pFGC5941, and the *ScATG8* sense sequence was ligated into the digested pFGC5941 vector to generate the pFGC5941-ScATG8-Sense sequence. The antisense sequence of the *ScATG8* gene and the pFGC5941-ScATG8-Sense vector were then digested using the restriction enzymes PacI and BamHI. The antisense sequence was ligated into the digested pFGC5941-ScATG8-Sense vector to generate the pFGC5941-ScATG8-silencing vector for RNA interference in *S. caninervis*.

### 4.6. S. caninervis Transformation

Agrobacterium tumefaciens (carrying pCABIM1300-CFP, pCAMBIA1300-CFP-ScATG8, pFGC5941, and pFGC5941-ScATG8 vectors) stored at −80 °C were activated on LB solid medium, and single clones were picked for shaking. The culture was incubated overnight until OD600 = 0.6 and transferred to AB:MES (17.2 mM K_2_HPO_4_, 8.3 mM NaH_2_PO_4_, 18.7 mM NH_4_Cl, 2 mM KCl, 1.25 mM MgSO_4_, 100 μM CaCl_2_, 10 μM FeSO_4_, 50 mM MES, 2% glucose (*w*/*v*), 200 μM acetosyringone) for resuspension and overnight incubation (28 °C, 200 rpm). The bacterial culture was centrifuged at 5000 rpm for 10 min, and the collected bacteria were suspended in liquid medium (AB:MES:knop (*v*/*v*) = 1:1) until OD600 = 0.8. We then infiltrated the prepared bacterial solution with cultured *S. caninervis*, infested at 28 °C for 50 rpm for 3 h (with fresh bacterial solution changed every hour), and then placed it on a substrate soil block to grow.

### 4.7. Generation of Transgenic P. patens

*P. patens* cultured for 7 d was put into pCAMBIA1300-CFP-ScATG8 infiltration solution for 1 h, and then the excess infiltration solution was sucked out and put into a light incubator for incubation. After one week of transformation, *P. patens* was screened with 50 mg/L thaumatin until overexpression-positive plants appeared, and a single gametophyte was grown and propagated for expansion to a single line.

### 4.8. Confocal Microscopy of Leaves in N. benthamiana and S. caninervis

Tobacco and *S. caninervis* were transiently transformed via Agrobacterium injection and transfection, respectively. After 36 h of transient transformation, leaves around the injection site of tobacco were taken with a punch, and green leaf tissues of *S. caninervis* were pressed with forceps. The localization of ScATG8 in the subcellular matrix and autophagy were observed using a laser confocal microscope (LSM800, Oberkochen, Germany).

### 4.9. Measurement of Fv/Fm (Maximum Quantum Yield of PS II)

The experiment was conducted using a pulsed modulated fluorometer (PAM-2500, manufactured by Heinz Walz). *S. caninervis* was dark treated for at least half an hour before measurements were conducted under shade. During dehydration–rehydration, we measured time points at 0, 2 h, 6 h, 8 h, 12 h, 20 h, 24 h, and 48 h, and during rehydration, we measured time points at 0, 2 h, 6 h, 12 h, and 24 h. The maximum quantum yield of PS II (0.1 μmol m^−2^ s^−1^), i.e., Fv/Fm. The measurement of photoexcitation and production of a minimum fluorescence yield (F0), as well as saturating pulses (8000 μmol m^−2^ s^−1^), after 0.8 s turned off all reaction centers and produced a maximum fluorescence yield (Fm). Variable chlorophyll fluorescence (Fv) = Fm − F0 [33].

### 4.10. Measurement of Absolute Water Content (AWC)

Samples of *S. caninervis* were taken and weighed as fresh weight (Fw) at 0, 6, 24, and 48 h after dehydration treatment, followed by immediate rehydration and fresh weight at 0.5, 6, and 24 h. We put the samples in the oven until all the water was removed and weighed the dry-weight samples (Dw). Finally, the results were calculated as follows: AWC = (Fw − Dw)/Dw × 100% [36].

### 4.11. NBT and DAB Staining

DAB and NBT staining solutions were used to detect the active sites of peroxidase in the cells; the concentrations were 1 mg/mL and 0.5 mg/mL, respectively. DAB needs to adjust the PH = 5.8 and add 0.1% Tritonx-100 and keep it away from light. The *S. caninervis* samples were taken at 0 h, 6 h, 24 h, and 48 h of dehydration and 0.5 h, 6 h, and 24 h of hydration, respectively, before being put into DAB and NBT staining solution and stained for 6–8 h after vacuum extraction for 10 min. Subsequently, we poured off the staining solution, added 95% ethanol, and boiled the samples in a 95% water bath for 10 min. Finally, the samples were immersed in 95% ethanol and decolorized at room temperature until the decolorized solution was clear.

### 4.12. Statistical Analysis

For the statistical analysis, the one-way ANOVA test was performed using SPSS 25. The level of significance was set at *p* < 0.05. The data were visualized using GraphPad prism 8.0.2.

## Figures and Tables

**Figure 1 plants-13-00059-f001:**
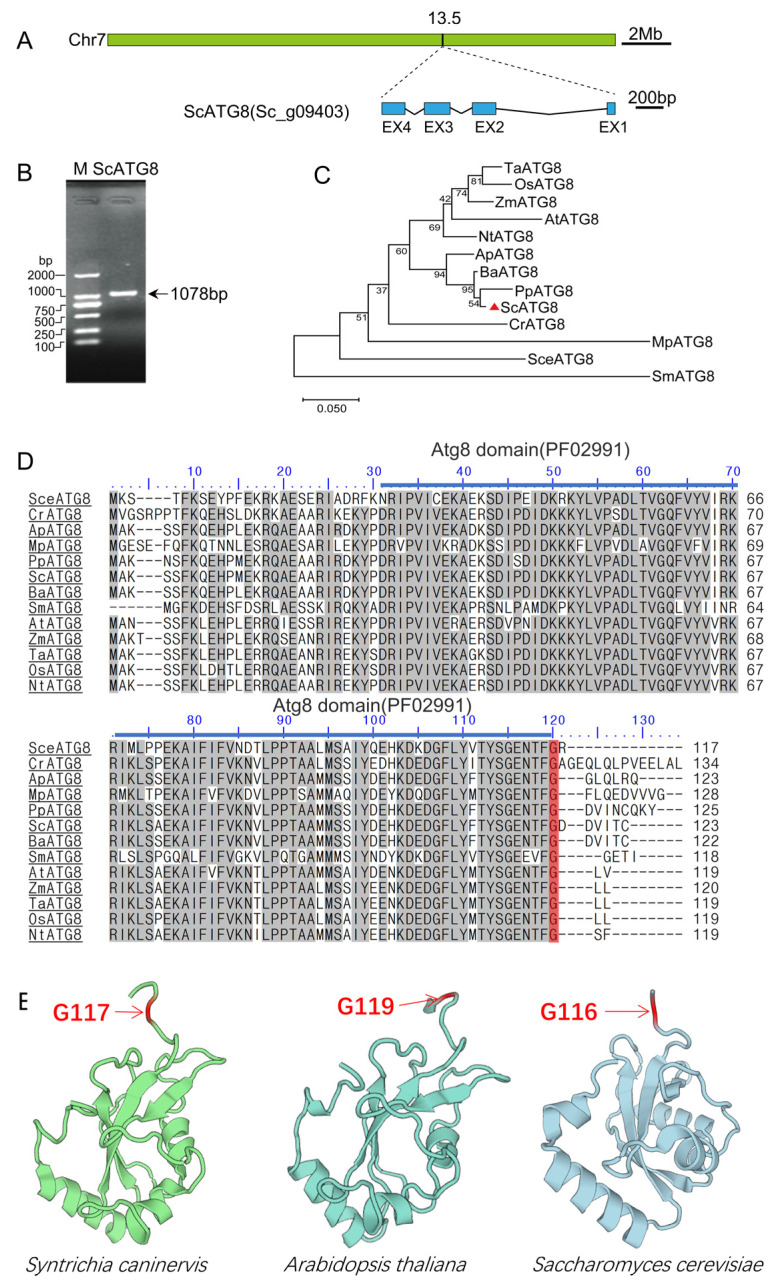
Structural and phylogenetic characterization of the ScATG8 protein. (**A**) Schematic presentation of the chromosomal location and structure of *ScATG8* gene. (**B**) PCR cloning of the *ScATG8* gene; (**C**) phylogenetic tree of ATG proteins from different species. The full lengths of the amino acid sequences of ATG8 proteins were aligned using the Clustal W method in MEGAX. An unrooted phylogenetic tree was built using the Maximum Likelihood method with 1000 replicates. ScATG8 was marked by a red triangle. The numbers at the nodes represent statistical frequency. The protein sequences used for the phylogeny are provided in Appendix A. (**D**) Multiple sequence alignment of ATG8 proteins from different species. Amino acid sequences were aligned using Clustal W and visualized using BioEdit. The ATG8 functional domain was indicated with blue-colored lines. The protein sequences used for the sequence alignment were provided in Appendix A. (**E**) The 3D structure of ATG8 proteins. The 3D models were generated using SWISS-MODEL [26]. The helix shape represents an α helix, the arrow shape represents a β sheet, the arrowhead represents the carboxyl termini of β sheets. The C-terminal cleavage glycine (G) site is shown in red.

**Figure 2 plants-13-00059-f002:**
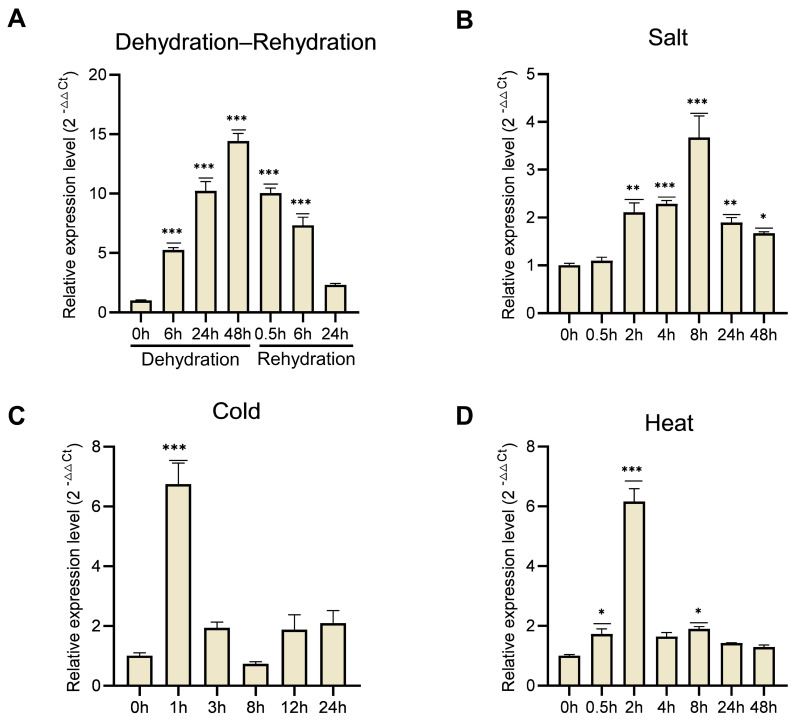
RT-qPCR data analysis in different stress conditions. RT-qPCR verification of the *ScATG8* gene during dehydration–rehydration (**A**), salt (**B**), cold (**C**), and heat (**D**) stresses. For the dehydration–rehydration process, the gametophytes were slowly dried (25 °C, 67%) over 48 h (D0 h, 6 h, 24 h and 48 h), followed by immediate rehydration for 24 h (R0.5 h, 6 h, 24 h). For salt stress, the gametophytes were exposed to 250 mM NaCl for 48 h. For cold stress, the gametophytes were growth at −4 °C for 24 h; for heat stress, the gametophytes were growth at 40 °C for 3 h. *ScTubulin2* was chosen as the reference gene. The relative expression level of the target gene was calculated according to the 2^−ΔΔCt^ method [27], using 0 h as the control and setting its fold change to 1. Bars are shown as the standard deviation. There were three biological replicates for each set of determinations, and values are expressed as means of *n* = 3 ± SD; the significance was determined via ANOVA (* *p* < 0.05, ** *p* < 0.01, *** *p* < 0.001).

**Figure 3 plants-13-00059-f003:**
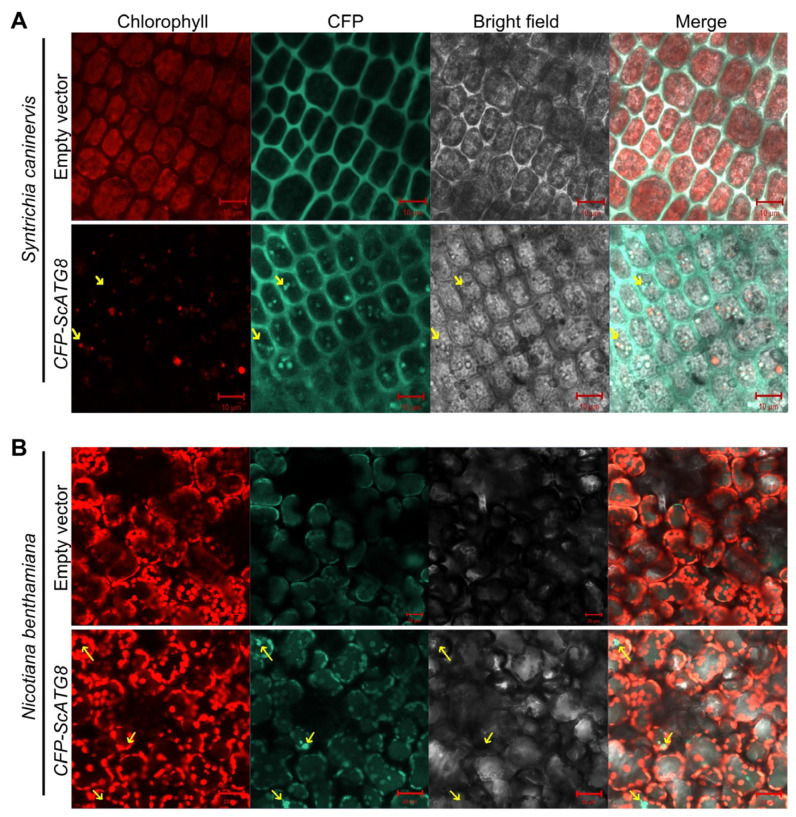
Subcellular localization of ScATG8 in *S. caninervis* and *Nicotiana benthamiana* cells under confocal microscopy. (**A**) CFP-ScATG8 transiently expressed *S. caninervis* cells. Bar = 10 μm. (**B**) CFP-ScATG8 transiently expressed *N. benthamina* mesophyll cells. Bar = 20 μm. Arrowheads indicate autophagosomes. At least three transiently expressing cells were photographed for the 35S::ScATG8 and empty vector (35s::CFP) groups.

**Figure 4 plants-13-00059-f004:**
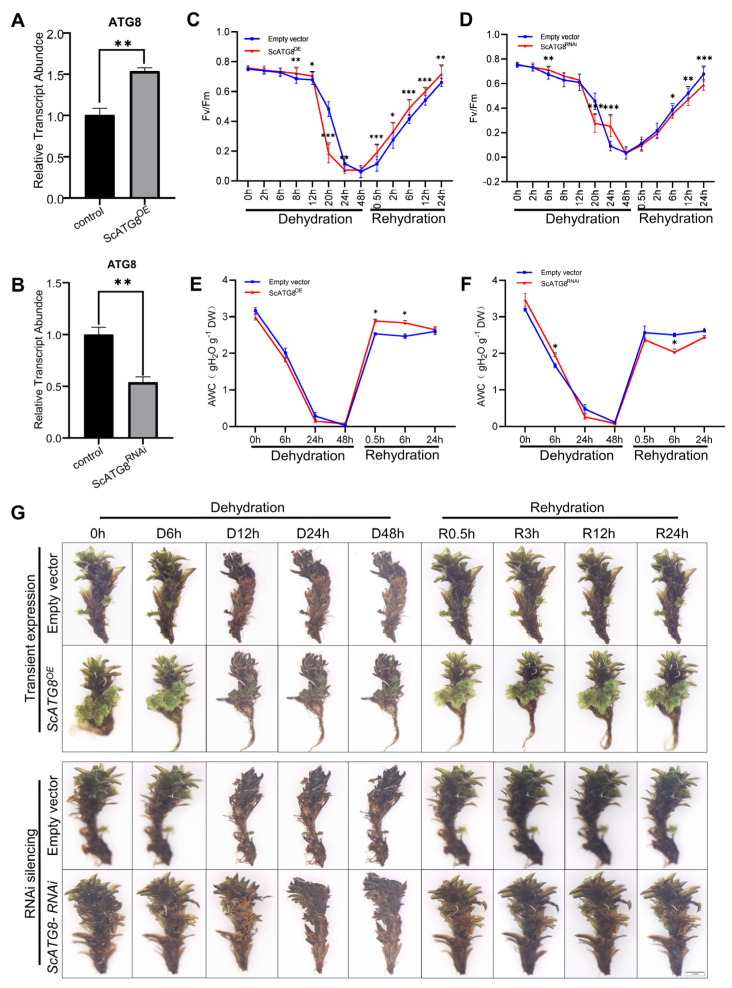
Impact of *ScATG8* gene expression on the response of *S. caninervis* to the dehydration–rehydration process. RT-qPCR identification of *S. caninervis* ScATG8^OE^ (**A**) and ScATG8^RNAi^ (**B**) plants. Fv/Fm measurements for ScATG8^OE^ (**C**) and ScATG8^RNAi^ (**D**) plants. Absolute water contents in ScATG8^OE^ (**E**) and ScATG8^RNAi^ (**F**) plants. (**G**) Morphological changes in ScATG8^OE^ and ScATG8^RNAi^ plants. Bar = 1 mm. There were three biological replicates for each set of determinations, and values are expressed as means of *n* = 3 ± SD; the significance was determined via ANOVA (* *p* < 0.05, ** *p* < 0.01, *** *p* < 0.001).

**Figure 5 plants-13-00059-f005:**
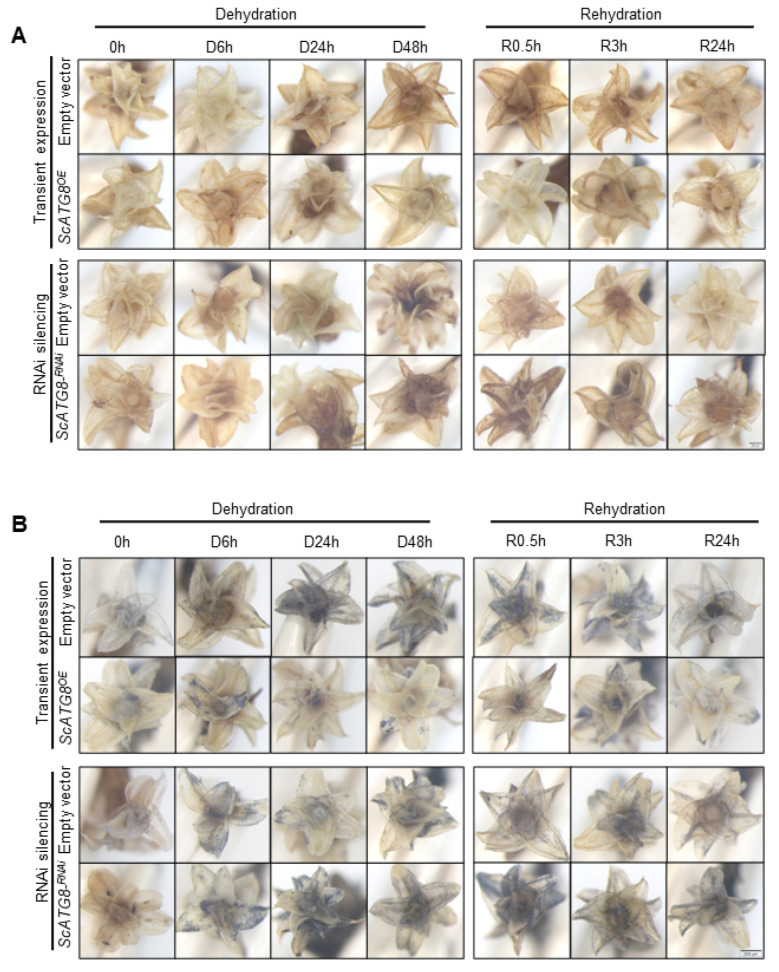
Impact of *ScATG8* gene expression on ROS level in *S. caninervis*. (**A**) DAB staining of ScATG8^OE^ and ScATG8^RNAi^ plants. Bar = 20 μm. (**B**) NBT staining of ScATG8^OE^ and ScATG8^RNAi^ plants. Bar = 20 μm.

**Figure 6 plants-13-00059-f006:**
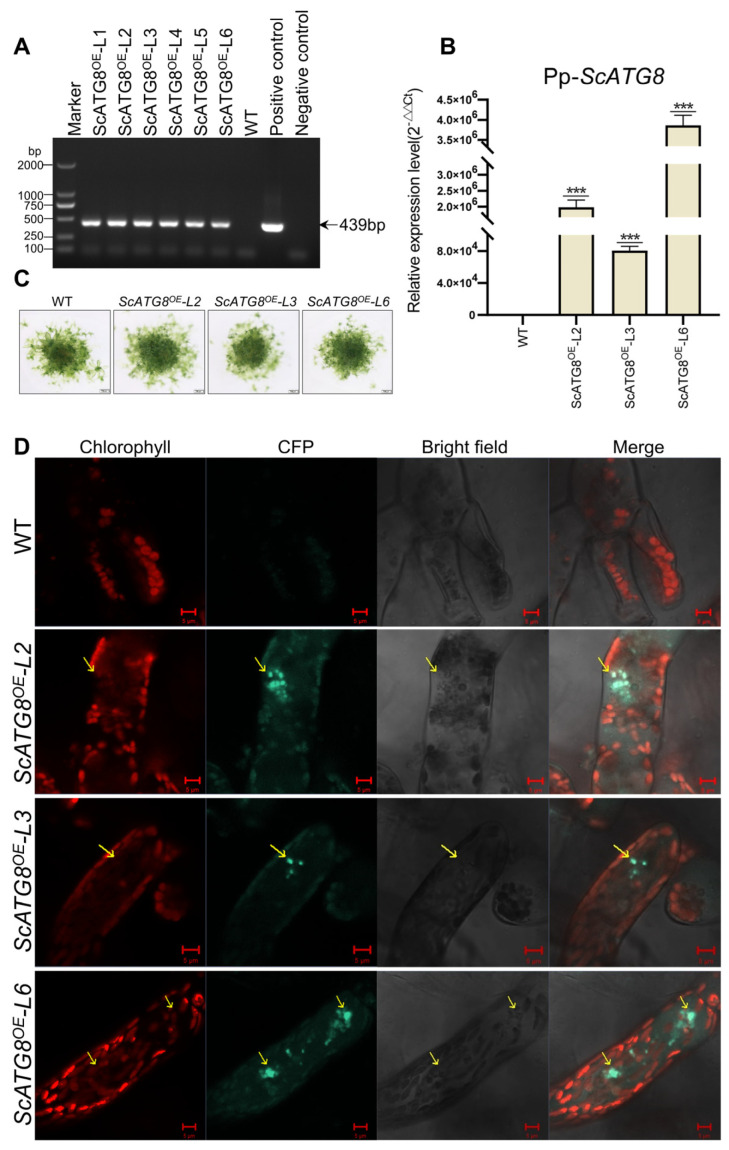
Growth of *ScATG8* transgenic *P. patens* under different stress conditions. (**A**) Determination of transgenic *P. patens* via PCR. Sequence specific primers were used to clone the *ScATG8* from transgenic *P. patens* lines. (**B**) RT-qPCR for the detection of *ScATG8* expression in transgenic *P. patens* lines. There were three biological replicates for each set of determinations, and values are expressed as means of *n* = 3 ± SD; the significance was determined via ANOVA (*** *p* < 0.001). (**C**) Phenotype of the ScATG8^OE^ transgenic *P. patens*. Bar = 500 μm. Statistics on the number. (**D**) CFP-ScATG8 transgenic *P. patens* lines’ protofilament. Bar = 5 μm. Arrowheads indicate autophagosomes. At least three transgenic-expressing cells were photographed for the 35S::ScATG8 groups.

**Figure 7 plants-13-00059-f007:**
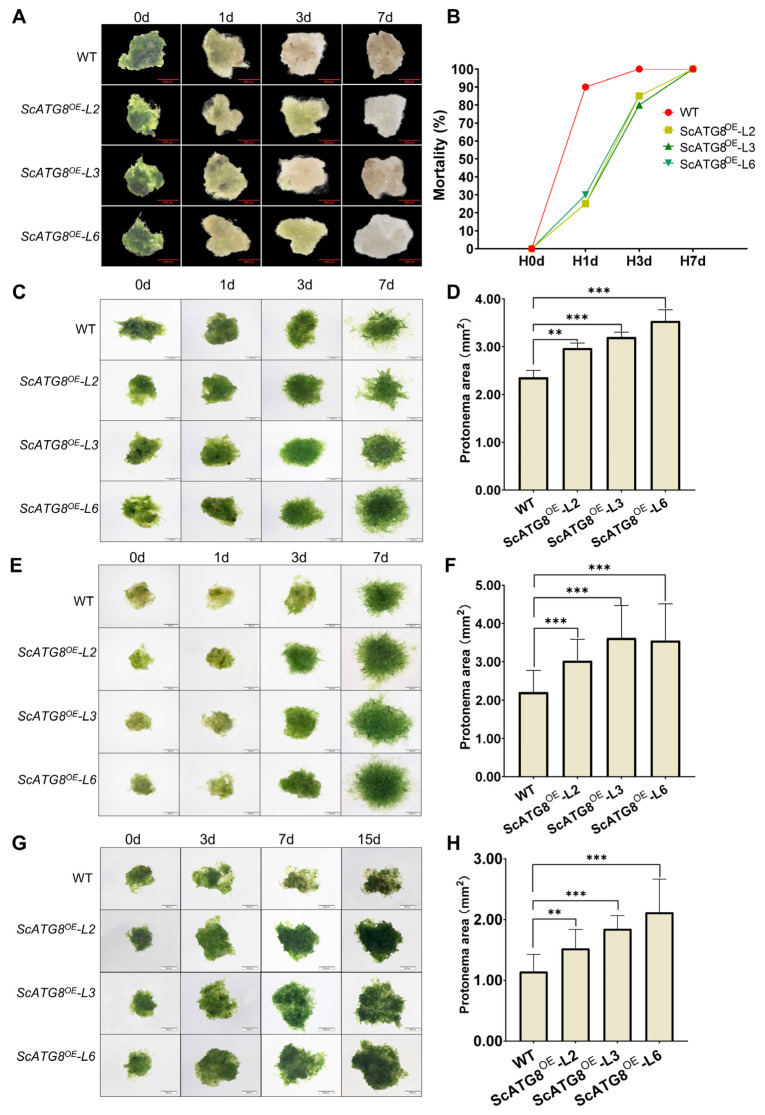
Growth of *ScATG8* transgenic *P. patens* under different stress conditions. (**A**) Survival of ScATG8^OE^ transgenic *P. patens* under desiccation. The protonemal colonies of ScATG8^OE^ transgenic *P. patens* were dehydrated for 24 h and immediately rehydrated and cultured using Knop’s agar medium for 7 d. Photos were taken on days 0, 1, 3, and 7 after hydration. Bar = 500 μm. (**B**) The mortality rate in (**A**) was defined as the ratio of the number of colonies that died on 0 d, 1 d, 3 d, and 7 d. A total of 20 colonies were used for the statistics. Growth of ScATG8^OE^ transgenic *P. patens* under heat stress (**C**), cold stress (**E**), and salt stress (**G**). Photos were taken on days 0, 1, 3, and 7 after heat and cold stress treatment and days 0, 3, 7, and 15 after the salt stress treatment. Bar = 500 μm. The protonemal area was calculated after 7 d of heat stress (**D**), 7 d of cold stress (**F**), and 15 d of salt stress treatments. (**H**) For each treatment, a total of 20 colonies were used. Values were presented as the means of *n* = 20 ± SD; significance was determined via ANOVA (** *p* < 0.01, *** *p* < 0.001).

## Data Availability

The data that support the findings of this study are presented in the article.

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
