# Peer review of "ScATG8 Gene Cloned from Desert Moss Syntrichia caninervis Exhibits Multiple Stress Tolerance"

_plants, 2023, doi:10.3390/plants13010059_

Round 1

Reviewer 1 Report

Comments and Suggestions for Authors

Some comments are shown in the attached file. 

Author Response

  1. Line 30 “Under desiccation stress, transient expression of ScATG8 enhanced the tolerance of S. caninervis and its ability to scavenge ROS”. “desiccation”:Do you mean drought or water deficit stress? What ROSs do you mean?

A: Desiccation, different from drought stress, refers severe to complete water loss, and leads lethal damage in most plant species. Here, ROS refers to H2O2 and O2. In our study we detected the H2O2 and O2 by DAB and NBT staining, respectively.

  1. Line 47 “accumulation of reactive oxygen species”: reduce accumulation of ROS?

and the next sentence you tell enhance the activity of SOD and ....! there are different sentences for one topic!

A: In the introduction we mentioned “reduces photosystem II damage and the accumulation of reactive oxygen species”. This is the result obtained by Yue et al through experiments. Please refer to original research paper.

Reference:

Yue, J.Y.; Wang, W.W.; Dou, X.T.; Wang, Y.J.; Jiao, J.L.; Wang, H.Z. Overexpression of the autophagy-related gene TaATG8 enhances wheat seedling tolerance to salt stress by increasing autophagic activity. Crop Pasture Sci. 2022, 73, 1325-1333, doi:10.1071/cp22086.

  1. scientific names and gene names should be shown in Italic form. Check whole text.

A: Thanks for your suggestion. We revised part as you suggested in our manuscript.

  1. Line 234 : “[26]” No subscript!

A: Thanks for your suggestion. We revised this part as you suggested in our manuscript.

  1. Line 278-282: “space...” Please check other typo errors.

A: Thanks for your suggestion. We revised this part as you suggested in our manuscript.

Reviewer 2 Report

Comments and Suggestions for Authors

Authors have attempted to exploit the ATG8 gene from Syntrichia caninervis, a plant that can survive desiccated environmental conditions, in the moss Physcomitrella patens to study its function under multiple abiotic stresses. Study looks interesting and designed appropriately. However authors have not justified why they have chosen Physcomitrella patens for the study, and introduction has failed to indicate the same. Moreover, the moss Physcomitrella patens has shown higher tolerance to different abiotic stress conditions salt, osmotic and dehydration stress (Frank et al. 2005 Planta). It’s a good model system, no doubt, but could have been worthwhile to study the function in a crop plant to improve tolerance.

Further authors have used desiccation and dehydration frequently in similar context. But, there is a physiological distinction between these two. Zhang & Bartels (2018 Jour Exp Bot) state that “ dehydration implies that whole plants or detached organs encounter a steady water loss and are often kept in air to lose water; and (iii) desiccation is the final result of dehydration and the water status is equilibrated with the air (desiccated is equal to extremely dehydrated)”.  I suggest that authors should look into this and consider the terminology.

Authors mentioned in the introduction (line 41) ‘in plants, ATG8 has been found to be involved in a wide range of biotic and abiotic stresses [3]’. If so, why authors have again attempted to study ‘novel role of the gene in response to multiple abiotic stresses.’ 

Authors mentioned (line 40): Salt stress, cold stress and heat stress were 284 applied by incubating S. caninervis gametophytes in 250 mM NaCl solution, 4 °C and 285 °C. On what basis were these values selected; any references available or authors own work. More detailed account of methodology adopted will be better.

Results on ‘Expression pattern of ScATG8 gene under different abiotic stresses’ have not written in detail any other abiotic stresses other than dehydration.

Lines 187-189: A large number of wild type plants died after desiccation treatment on the first day, whereas transgenic plants  survived until the third day of stress treatment, but all plants died on the seventh day 189 (Figure 7A, B).  Does it mean that no transgenic plants survived after a week?  This is a bit surprising to see considering the abiotic stress tolerance nature of  P. patens. Can authors discuss. Is it only functional during the early stages of stress?  

Comments on the Quality of English Language

moderate revision will be good

Author Response

  1. Authors have attempted to exploit the ATG8 gene from Syntrichia caninervis, a plant that can survive desiccated environmental conditions, in the moss Physcomitrella patens to study its function under multiple abiotic stresses. Study looks interesting and designed appropriately. However, authors have not justified why they have chosen Physcomitrella patens for the study, and introduction has failed to indicate the same. Moreover, the moss Physcomitrella patens has shown higher tolerance to different abiotic stress conditions salt, osmotic and dehydration stress (Frank et al. 2005 Planta). It’s a good model system, no doubt, but could have been worthwhile to study the function in a crop plant to improve tolerance.

A: Thank you for your comments. As you suggested we have revised the introduction. The reason why we chose Physcomitrella patens as the research material is as follows. (1) The Physcomitrella patens and Syntrichia caninervis are evolutionarily close. In addition, the genomes of these two species shared high collinearity and shared common proteins [1,2].

(2) The transgenic screening approaches for Physcomitrella patens is relatively mature, efficient, and timesaving.

(3) Compared to Syntrichia caninervis, the Physcomitrella patens is relatively sensitive to dry desiccation stress. P. patens relatively sensitive to desiccation stress than S. caninervis, P. patens did not survive water potentials < −13 MPa [3] while S. caninervis withstand water potentials of -540 MPa [4].Thus, Physcomitrella patens is the best model to study the function of genes from Syntrichia caninervis.

We fully agree with you that Physcomitrella patens is a model plant, and also transforming of model crops such as rice or cotton has greater significance. However, we used Physcomitrella patens due to is the best candidate for studying desiccation tolerance genes from Syntrichia caninervis.We are also attempting to transfer some resistance genes from Syntrichia caninervis into crops, and if there is a possibility in the future, we will also transform the ScATG8 gene into model plants or crops.

References:

[1] Silva, A.T., Gao, B., Fisher, K.M., Mishler, B.D., Ekwealor, J.T.B., Stark, L.R., Li, X., Zhang, D., Bowker, M.A., Brinda, J.C., Coe, K.K. and Oliver, M.J. (2021), To dry perchance to live: Insights from the genome of the desiccation-tolerant biocrust moss Syntrichia caninervis. Plant J, 105: 1339-1356. https://doi.org/10.1111/tpj.15116

[2] Gao, B., Zhang, D., Li, X. et al. De novo assembly and characterization of the transcriptome in the desiccation-tolerant moss Syntrichia caninervis. BMC Res Notes 7, 490 (2014). https://doi.org/10.1186/1756-0500-7-490.

[3] Koster, K.L., Balsamo, R.A., Espinoza, C. et al. Desiccation sensitivity and tolerance in the moss Physcomitrella patens: assessing limits and damage. Plant Growth Regul 62, 293–302 (2010). https://doi.org/10.1007/s10725-010-9490-9

[4] Oliver MJ, Velten J, Mishler BD. Desiccation tolerance in bryophytes: a reflection of the primitive strategy for plant survival in dehydrating habitats?. Integr Comp Biol. 2005;45(5):788-799. doi:10.1093/icb/45.5.788

  1. Further authors have used desiccation and dehydration frequently in similar context. But, there is a physiological distinction between these two. Zhang & Bartels (2018 Jour Exp Bot) state that “ dehydration implies that whole plants or detached organs encounter a steady water loss and are often kept in air to lose water; and (iii) desiccation is the final result of dehydration and the water status is equilibrated with the air (desiccated is equal to extremely dehydrated)”. I suggest that authors should look into this and consider the terminology.

A: We completely agree with your views and are very grateful for your suggestions. We revised as you suggested.

  1. Authors mentioned in the introduction (line 41) ‘in plants, ATG8 has been found to be involved in a wide range of biotic and abiotic stresses [3]’. If so, why authors have again attempted to study ‘novel role of the gene in response to multiple abiotic stresses.’

A: Although it has been reported that ATG8 genes mediate many abiotic stress response processes, there has been no report on ATG8 mediating desiccation stress to date. In addition, there have been no reports on the function of ATG8 genes from Syntrichia caninervis as desiccation tolerance bryophytes. Therefore, here we are studying the function of ATG8 genes in bryophytes.

  1. Authors mentioned (line 40): Salt stress, cold stress and heat stress were applied by incubating S. caninervis gametophytes in 250 mM NaCl solution, -4 °C and 40°C. On what basis were these values selected; any references available or authors own work. More detailed account of methodology adopted will be better.

A: For the salt and heat stress treatment of S. caninervis gametophytes, we referred to the previous research [1,2]. For the cold stress treatment, we referred the unpublished data of S. caninervis tolerance to cold treatment.

Reference:

[1] Li, X.S.; Yang, R.R.; Liang, Y.Q.; Gao, B.; Li, S.M.; Bai, W.W.; Oliver, M.J.; Zhang, D.Y. The ScAPD1-like gene from the desert moss Syntrichia caninervis enhances resistance to Verticillium dahliae via phenylpropanoid gene regulation. Plant J. 2023, 113, 75-91, doi:10.1111/tpj.16035.

[2] Xu S J , Liu C J , Jiang P A ,et al.The effects of drying following heat shock exposure of the desert moss Syntrichia caninervis.[J].Science of the Total Environment, 2009, 407(7):2411-2419.DOI:10.1016/j.scitotenv.2008.12.005.

  1. Results on ‘Expression pattern of ScATG8 gene under different abiotic stresses’ have not written in detail any other abiotic stresses other than dehydration.

A: Thanks for your comment. We revised this part as you suggested in our manuscript.

  1. Lines 187-189: A large number of wild type plants died after desiccation treatment on the first day, whereas transgenic plants survived until the third day of stress treatment, but all plants died on the seventh day 189 (Figure 7A, B). Does it mean that no transgenic plants survived after a week? This is a bit surprising to see considering the abiotic stress tolerance nature of P. patens. Can authors discuss. Is it only functional during the early stages of stress?

A: In this study, for the desiccation of P. patens we used silica gel method which is applied by Mukae et al [1] previously, and a large number of wild type plants died after desiccation treatment on the first day. Transgenic plants do indeed die after one week and no transgenic plants survived. These results consistent with Mukae et al [1]. In addition to this, previous research reported that P. patens did not routinely survive water potentials < −13 MPa [2]. Therefore, the wild-type plants die after one day treatment is consistent with the above fact.

References:

[1] Mukae, K.; Sakil, M.A.; Kotake, T.; Inoue-Aono, Y.; Moriyasu, Y. Autophagy accelerates cell death after desiccation and hydration stress in Physcomitrium. Environ. Exp. Bot. 2023, 213, 12, doi:10.1016/j.envexpbot.2023.105412.

[2] Koster, K.L., Balsamo, R.A., Espinoza, C. et al. Desiccation sensitivity and tolerance in the moss Physcomitrella patens: assessing limits and damage. Plant Growth Regul 62, 293–302 (2010). https://doi.org/10.1007/s10725-010-9490-9

Reviewer 3 Report

Comments and Suggestions for Authors

This manuscript describes the isolation of ScATG8 gene from desert moss Syntrichia caninervis and functional characterization under several abiotic stress conditions. Authors present new research data but the manuscript must be improved and needs revision. The first concerns the analysis of the phenotypic growth of the three P. patens lines that overexpress the scATG8 gene under abiotic stress conditions. To support the data relating to the area of ​​the protonemal colonies it would be more decisive to introduce a table with the fresh and/or dry weights data of these lines and compare them with the control line by statistics. Moreover, the “Discussion” paragraph should be improved and also by evaluating/comparing the results obtained from the functional analysis of the ScATG8 gene under abiotic stresses with other research works (already cited in the introduction). Finally, I suggest the following minor revisions.

Minor corrections

- Lane 28: substitute “… ptent ...” with “… patent …”

- Lane 30: substitute “…, transient expression of ScATG8 enhanced …” with “…, overexpression of ScATG8 enhanced …”

- Lane 31: substitute “In addition, the transgene of ScATG8 promoted …” with “In addition, ScATG8 overexpression promoted …”

- Lane 77: substitute “Overexpression ScATG8 in P. patens …” with “Overexpression of ScATG8 gene in P. patens …”

- Lane 145: substitute “… ScATG8 transient expressed …” with “… ScATG8 overexpressed …”

- Lane 146: substitute “… gene silenced (ScATG8RNAi) (Figure 4B) …” with “… gene silenced (ScATG8RNAi) lines (Figure 4B) …”

- Lane 184: substitute “… conditions (Figure 6B).” with “… conditions (Figure 6C).”

- Lane 253: substitute “… plants experienced better …” with “… plants showed better …”

- Lane 256: substitute “In addition, the overexpression plants enhanced …” with “In addition, the ScATG8 overexpression enhanced …”

- Lanes 284 to 285: the authors have to introduce in this sentence the times in which the gametophytes were subjected to the different stresses: for 48 h (salt stress); for 24 h (cold stress); for 3 h (heat stress)

- Lane 286: the authors stated “Samples were taken at 0, 0.5, 2, 4, 8, 24, and 48 h, respectively.”. I’m not fully in agreement with this sentence. Looking at the Figure 2, samples were taken as above described for salt and heat stress but not for cold stress where samples were taken at 0,1,3,8,12 and 24 h

- Lane 338: substitute “The cultured Agrobacterium …” with “The bacterial culture …”

- Lanes 335 and 339: authors have to introduce near OD the wavelength in nm

- Lane 338: substitute “The cultured Agrobacterium …” with “The bacterial culture …”

- Lanes 344 to 350: authors need to improve this part of the manuscript by simplifying it and making it more understandable

- Lane 352: substitute “… transiently expressed by Agrobacterium …” with “… transiently transformed by Agrobacterium …”

Author Response

1 This manuscript describes the isolation of ScATG8 gene from desert moss Syntrichia caninervis and functional characterization under several abiotic stress conditions. Authors present new research data but the manuscript must be improved and needs revision. The first concerns the analysis of the phenotypic growth of the three P. patens lines that overexpress the ScATG8 gene under abiotic stress conditions. To support the data relating to the area of the protonemal colonies it would be more decisive to introduce a table with the fresh and/or dry weights data of these lines and compare them with the control line by statistics. Moreover, the “Discussion” paragraph should be improved and also by evaluating/comparing the results obtained from the functional analysis of the ScATG8 gene under abiotic stresses with other research works (already cited in the introduction). Finally, I suggest the following minor revisions.

A:Thank you very much for your valuable feedback. Based on your feedback, we have carefully revised the manuscript and improved the discussion part. In this study, we only measured the growth area, and it was difficult to determine the biomass of protonema due to is quantity is very low. In addition, we have also provided growth rate of transgenic lines in the revised manuscript.

2.Minor corrections

- Lane 28: substitute “… ptent ...” with “… patent …”

- Lane 30: substitute “…, transient expression of ScATG8 enhanced …” with “…, overexpression of ScATG8 enhanced …”

- Lane 31: substitute “In addition, the transgene of ScATG8 promoted …” with “In addition, ScATG8 overexpression promoted …”

- Lane 77: substitute “Overexpression ScATG8 in P. patens …” with “Overexpression of ScATG8 gene in P. patens …”

- Lane 145: substitute “… ScATG8 transient expressed …” with “… ScATG8 overexpressed …”

- Lane 146: substitute “… gene silenced (ScATG8RNAi) (Figure 4B) …” with “… gene silenced (ScATG8RNAi) lines (Figure 4B) …”

- Lane 184: substitute “… conditions (Figure 6B).” with “… conditions (Figure 6C).”

- Lane 253: substitute “… plants experienced better …” with “… plants showed better …”

- Lane 256: substitute “In addition, the overexpression plants enhanced …” with “In addition, the ScATG8 overexpression enhanced …”

- Lanes 284 to 285: the authors have to introduce in this sentence the times in which the gametophytes were subjected to the different stresses: for 48 h (salt stress); for 24 h (cold stress); for 3 h (heat stress)

- Lane 286: the authors stated “Samples were taken at 0, 0.5, 2, 4, 8, 24, and 48 h, respectively.”. I’m not fully in agreement with this sentence. Looking at the Figure 2, samples were taken as above described for salt and heat stress but not for cold stress where samples were taken at 0,1,3,8,12 and 24 h

- Lane 338: substitute “The cultured Agrobacterium …” with “The bacterial culture …”

- Lanes 335 and 339: authors have to introduce near OD the wavelength in nm

- Lane 338: substitute “The cultured Agrobacterium …” with “The bacterial culture …”

- Lanes 344 to 350: authors need to improve this part of the manuscript by simplifying it and making it more understandable

- Lane 352: substitute “… transiently expressed by Agrobacterium …” with “… transiently transformed by Agrobacterium …”

A: Thanks for your suggestions. We have carefully revised every point you raised.

Reviewer 4 Report

Comments and Suggestions for Authors

Comments on the Quality of English Language

The writing is good.

Author Response

  1. This study by Cao et al. describes that ScATG8 gene cloned from desert moss Syntrichia caninervis enhances multiple stress tolerance ability. They cloned ScATG8 gene encoded a protein with highly conserved 28 ATG8 functional domain and found that ScATG8 gene was increasingly expressed under different abiotic stresses. The transgene of ScATG8 promoted the growth of P. patens under multiple stress conditions. Those results document that ScATG8 may be a multifunctional gene, and plays a criticalrole in the survival of S. caninervis under various abiotic stresses. The topic is interesting, and the manuscript is well-writing and organized. However, There are some key points to be confirmed before publishing.
  2. Results 2.2 interpretation of Fig. 2B, 2C, and 2D in the text.

A: Thanks for your suggestion. We revised part as you suggested in our manuscript.

  1. Results 2.6 lines and wild type (WT) under normal growth conditions (Figure 6B). should be Figure 6C.

A: Thanks for your suggestion. We revised part as you suggested in our manuscript.

  1. Fig.2 legend. Add name of reference gene.

A: Thanks for your suggestion. We add the name of reference gene in the manuscript as you requested.

  1. Fig. 6 legend. (c) Statistics on the number. No data to show data;

A: Thanks for your suggestion. Fig. 6 legend. (c) Mainly comparing the growth status between WT and transgenic plants is to compare whether there is any difference between WT and transgenic plants. Thus we don’t provide any data.

  1. Add supplemental data for primer information including the positive control.

A: Thanks for your suggestion. The cDNA from wild type Syntrichia caninervis was used as a positive control to confirm transgenic P. patens. The primers of ScATG8 gene used for the PCR test and listed in Supplemental Table 1.

  1. Line 58. Delete the comma after plants.

A: Thanks for your suggestion. We revised part as you suggested in our manuscript.

  1. Line 77. P. patens should be in italic type. So is Line 160 S. caninervis and Line 182. Delete the comma after study.

A: Thanks for your suggestion. We revised part as you suggested in our manuscript.